# Screening of potential regulatory genes in carotid atherosclerosis vascular immune microenvironment

Yi Zhang[1], Lingmin Zhang[2], Yunfang Jia[3], Jing Fang[4], Shuancheng Zhang[4‡], Xianming Hou[1‡]*

1 Heibei Key Laboratory of Chinese Medicine Research on Cardio-cerebrovascular Disease, Hebei University of Traditional Chinese Medicine, Shijiazhuang City, Hebei Province, China, 2 Teaching and Research Office of Typhoon Fever Theory at the School of Basic Medicine, Hebei University of Traditional Chinese Medicine, Shijiazhuang City, Hebei Province, China, 3 Teaching and Research Office of Traditional Chinese Medicine History and Literature at the School of Basic Medicine, Hebei University of Traditional Chinese Medicine, Shijiazhuang City, Hebei Province, China, 4 Teaching and Research Office of Internal Canon of Medicine at the School of Basic Medicine, Hebei University of Traditional Chinese Medicine, Shijiazhuang City, Hebei Province, China

☯ These authors contributed equally to this work.
‡ SZ and XH also contributed equally to this work.
* xianminghou_201@163.com

**Data Availability Statement:** All RNA sequencing data files are available from the Gene Expression Omnibus (GEO) database (GEO accession number: GSE43292 and GSE28829).

## Abstract

### Background

Immune microenvironment is one of the essential characteristics of carotid atherosclerosis (CAS), which cannot be reversed by drug therapy alone. Thus, there is a pressing need to develop novel immunoregulatory strategies to delay this pathological process that drives cardiovascular-related diseases. This study aimed to detect changes in the immune micro-environment of vascular tissues at various stages of carotid atherosclerosis, as well as cluster and stratify vascular tissue samples based on the infiltration levels of immune cell subtypes to distinguish immune phenotypes and identify potential hub genes regulating the immune microenvironment of carotid atherosclerosis.

### Materials and methods

RNA sequencing datasets for CAS vascular tissue and healthy vascular tissue (GSE43292 and GSE28829) were downloaded from the Gene Expression Omnibus (GEO) database. To begin, the immune cell subtype infiltration level of all samples in both GSE43292 and GSE28829 cohorts was assessed using the ssGSEA algorithm. Following this, consensus clustering was performed to stratify CAS samples into different clusters. Finally, hub genes were identified using the maximum neighborhood component algorithm based on the construction of interaction networks, and their diagnostic efficiency was evaluated.

### Results

Compared to the controls, a higher number of immune cell subtypes were enriched in CAS samples with higher immune scores in the GSE43292 cohort. Advanced CAS was

**Funding:** This work was supported by Hebei Provincial Administration of Traditional Chinese Medicine Project, No: Z2022004. This type of funding mainly includes project funding, equipment procurement, and personnel support.

**Competing interests:** The authors declare that they have no competing interests.

characterized by high immune cell infiltration, whereas early CAS was characterized by low immune cell infiltration in the GSE28829 cohort. Moreover, CAS progression may be related to the immune response pathway. Biological processes associated with muscle cell development may impede the progression of CAS. Finally, the hub genes *PTPRC, ACTN2, ACTC1, LDB3, MYOZ2, and TPM2* had satisfactory efficacy in the diagnosis and prediction of high and low immune cell infiltration in CAS and distinguishing between early and advanced CAS samples.

## Conclusion

The enrichment of immune cells in vascular tissues is a primary factor driving pathological changes in CAS. Additionally, CAS progression may be related to the immune response pathway. Biological processes linked to muscle cell development may delay the progression of CAS. *PTPRC*, *ACTN2*, *ACTC1*, *LDB3*, *MYOZ2*, and *TPM2* may regulate the immune microenvironment of CAS and participate in the occurrence and progression of the disease.

## Introduction

As is well documented, carotid atherosclerosis (CAS), the pathological basis of carotid artery stenosis, is prevalent in middle-aged and elderly individuals [1]. According to a recent epidemiological survey, up to 12% of elderly men and 5% of women suffer from asymptomatic moderate atherosclerotic neck vessel stenosis, while severe carotid artery stenosis accounted for 3% and 1% of elderly men and women, respectively [2, 3]. Indeed, it has emerged as one of the leading causes of accidental death among the elderly population worldwide [4].

The carotid artery is frequently affected by atherosclerosis and is a major contributor to ischemic stroke [5–7]. The risk of cerebral ischemia in patients with carotid atherosclerosis depends on the degree of carotid artery stenosis [8–10]. The pathological basis of atherosclerosis chiefly involves lipid deposition under the intima of the middle great artery [11–13].

The immune response exerts a significant effect on the development of carotid atherosclerosis at all stages [14, 15]. A large number of immune factors were detected in carotid atherosclerotic lesions, including immune cells and their cytokines. Pro-inflammatory cytokines, such as TNF-α, IL-1β, and IL-12, may accelerate the progression of carotid atherosclerosis [16–19]. Conversely, anti-inflammatory cytokines such as IL-10, TGF-β, and Arg-1 may delay the development of carotid atherosclerosis [20, 21].

Immune microenvironment is one of the essential characteristics of carotid atherosclerosis. Given that carotid atherosclerosis cannot be reversed by drug therapy alone, there is an urgent need to formulate new immunoregulatory strategies to prevent this pathological process that causes cardiovascular-related diseases. Therefore, this study aimed to detect changes in the immune microenvironment of vascular tissues at different stages of carotid atherosclerosis, as well as cluster and stratify vascular tissue samples based on the infiltration levels of immune cell subtypes to distinguish between different immune phenotypes and identify potential hub genes regulating the immune microenvironment of carotid atherosclerosis.

## Materials and methods

### Data download

RNA sequencing datasets for CAS vascular tissue and healthy vascular tissue (GSE43292 [22] and GSE28829 [23]) were retrieved from the Gene Expression Omnibus (GEO) database.

Thirty-two CAS samples in the GSE43292 cohort were regarded as the CAS group, and paired 32 normal vascular tissue samples were designated as the control group. In the GSE28829 cohort, 13 early CAS samples and 16 advanced CAS samples were categorized as the early and advanced CAS groups.

## Quantification of immune microenvironment

The ssGSEA algorithm was used to convert RNA sequencing data from all samples to immune cell subtype infiltration data. Then, the immune scores of all samples were calculated using the ESTIMATE algorithm.

## Consensus cluster

Ward's method was utilized for consensus clustering to divide CAS samples in the GSE43292 cohort. When K = x (when the samples were divided into x clusters), the cumulative distribution function was the flattest in the range of consensus index values ([0.1, 0.9]).

## Differentially expressed genes (DEGs)

DEGs were identified using the following criteria: genes with a logarithm of fold change exceeding 1 and an adjusted P value less than 0.05.

## Enrichment analysis

With human genes as the background, enrichment analysis of DEGs was conducted using R package (clusterProfiler) based on gene ontology (GO) and Kyoto encyclopedia of genes and genomes (KEGG) databases. KEGG pathways and GO terms were retained based on a screening condition of $q < 0.05$.

## Hub gene screening

Protein-protein interaction networks (PPI) of DEGs were generated using the STRING website (http://string-db.org). The CytoHubba plug-in of Cytoscape 3.2 software and the maximum neighborhood component algorithm were employed to screen hub genes among DEGs. The top 10 genes were retained.

## Principal component analysis (PCA)

A linear function that converted original variables to new variables and removed redundant information was constructed. Principal component 1 and principal component 2 were used to construct a two-dimensional plane to characterize the phenotype of the samples.

## Statistical analysis

Correlation analysis was conducted using the Spearman test. The receiver operator characteristic curve (ROC) was plotted to evaluate the diagnostic value of DEGs. Considering that the datasets used are publicly available, the requirement for ethics committee review was waived.

## Ethical approval

All data in this study are publicly available and did not require ethics committee review.

## Results

### Changes in the immune microenvironment in CAS

With the exception of Type 2 T helper cells, central memory CD8 T cells, central memory CD4 T cells, and effector memory CD8 T cells, the levels of infiltration of the majority of immune cell subtypes were higher in the CAS samples compared to the control samples in the GSE43292 cohort (P < 0.001, Fig 1A). Likewise, the immune score in CAS samples was higher than that in control samples in the GSE43292 cohort (P < 0.001, Fig 1B). In the GSE43292 cohort, PCA based on immune cell subtype infiltration showed that CAS and control samples were clustered in different regions in the coordinate system composed of principal component 1 as the X-axis and principal component 2 as the Y-axis (Fig 1C). Overall, these results exposed that control and CAS samples possessed different immune microenvironment characteristics, which may play a decisive role in disease development and could be used to differentiate between control and CAS samples.

Similarly, the infiltration levels of the majority of immune cell subtypes were higher in the advanced CAS samples than in the early CAS samples in the GSE28829 cohort (Fig 2A). As anticipated, the immune score of advanced CAS samples was higher than that of early CAS samples in the GSE43292 cohort (P < 0.001, Fig 2B). Additionally, PCA based on immune cell subtype infiltration demonstrated that advanced and early CAS samples were clustered in different regions in the coordinate system composed of principal component 1 as the X-axis and principal component 2 as the Y-axis (Fig 2C). The immune microenvironment is a key hallmark of disease and changes throughout the disease process. Describing the phenotypes of diseases based on the immune microenvironment can delineate the heterogeneity of CAS and offer new insights into its pathogenesis.

### Consensus clustering of the CAS samples

To further elucidate the characteristics of the immune microenvironment in CAS samples and explore the underlying mechanism, consensus clustering was carried out on CAS samples in the GSE43292 cohort. When K = 2 (samples were divided into two clusters), the cumulative distribution function exhibited optimal flatness in the range of consensus index values ([0.1, 0.9], Fig 3A). Next, CAS samples in the GSE43292 cohort were divided into 2 clusters, with cluster A comprising 19 samples and cluster B consisting of 13 samples (Fig 3B). The infiltration levels of most immune cell subtypes, with the exception of Type 2 T helper cells, central memory CD8 T cells, central memory CD4 T cells, and effector memory CD8 T cells, were significantly higher in Cluster A, followed by cluster B and control samples, respectively (P < 0.001, Fig 3C). Similarly, the immune score in cluster A was significantly higher compared to cluster B and the control samples (P < 0.001, Fig 3D). PCA based on immune cell subtype infiltration uncovered that cluster A and cluster B samples were clustered in different regions of the coordinate system composed of principal component 1 as the X-axis and principal component 2 as the Y-axis (Fig 3E). Cluster A was characterized by high immune cell infiltration levels, whereas Cluster B was characterized by low immune cell infiltration levels.

### Enrichment analysis and hub gene screening

Compared to the control group, immune-related KEGG pathway gene sets, such as Toll-like receptor signaling pathways, antigen processing and presentation, and intestinal immune network for IgA production, were enriched in cluster B of the GSE43292 cohort (Fig 4A). On the other hand, immune-related KEGG pathway genes, such as antigen processing and presentation, were enriched in cluster A (Fig 4B). To further explore the enrichment mechanism of

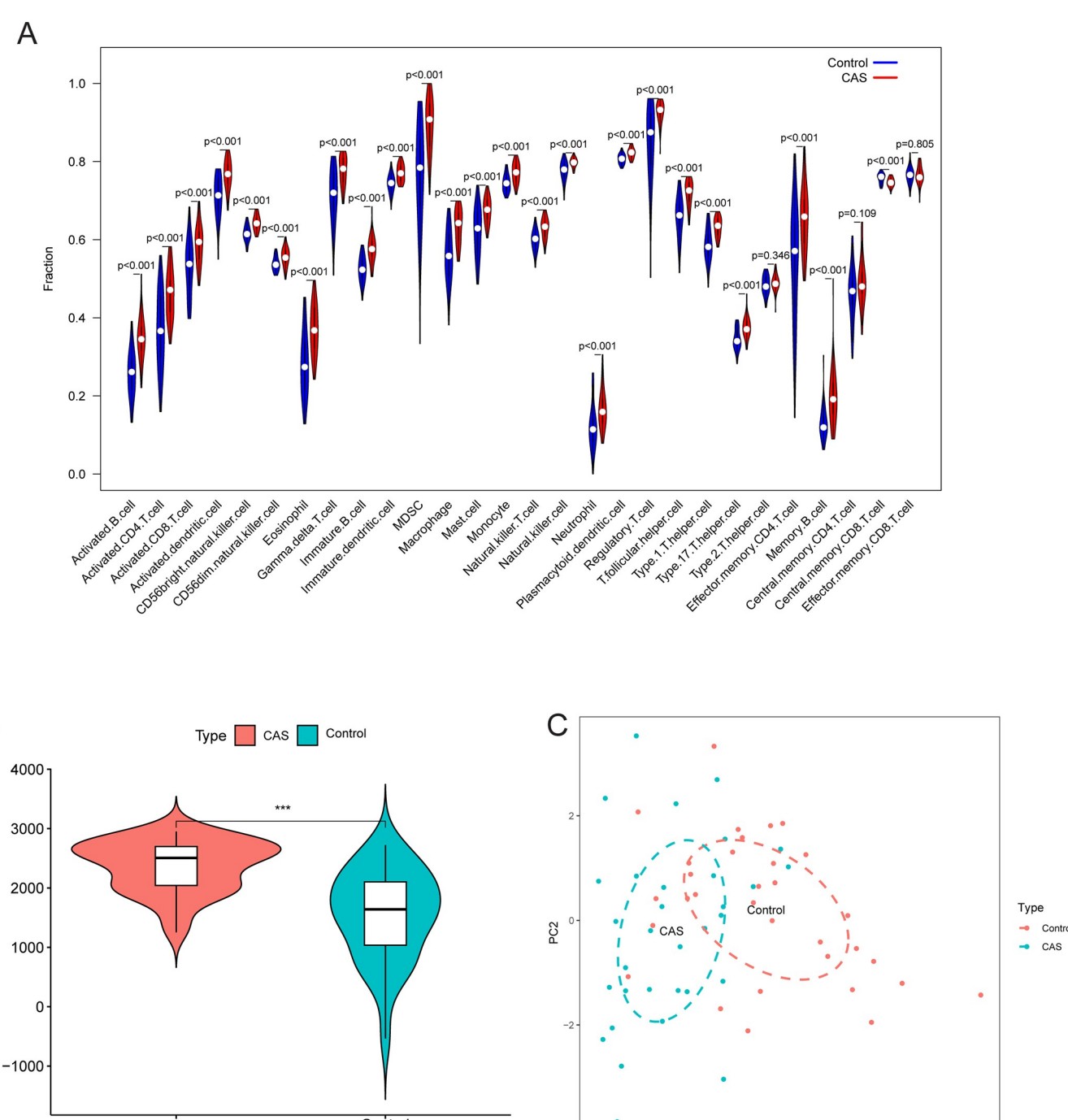

**Fig 1. Difference in the immune microenvironment between CAS and control samples in the GSE43292 cohort.** (A) Differences in immune cell subtypes between CAS and control samples. (B) Differences in immune scores between CAS and control samples. (C) Principal component analysis based on the infiltration level of immune cell subtypes in CAS and control samples. Abbreviation: CAS, carotid atherosclerosis; PC, principal component.

immune cells in CAS samples, DEGs in Cluster A and control samples of the GSE43292 cohort were investigated. Up-regulated DEGs in cluster A of the GSE43292 cohort were enriched in biological processes related to the activation of immune response, leukocyte migration, and

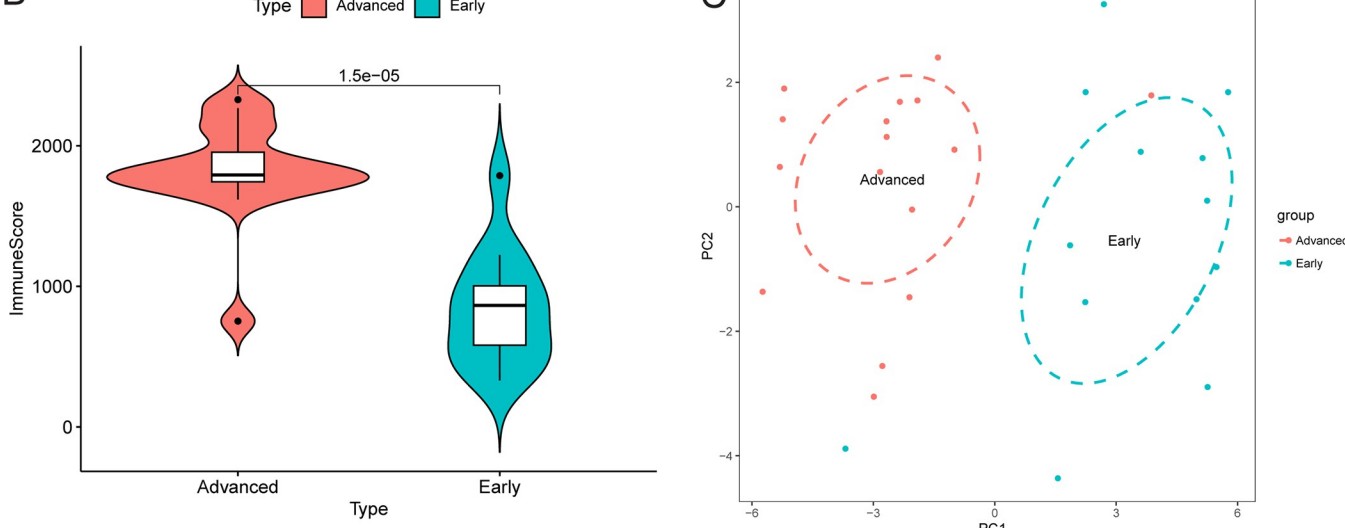

**Fig 2. Differences in the immune microenvironment between early and advanced carotid atherosclerosis samples in the GSE28829 cohort.** (A) Differences in immune cell subtypes between early and advanced CAS samples. (B) Differences in immune scores between early and advanced CAS samples. (C) Principal component analysis based on the infiltration level of immune cell subtypes in early and advanced CAS samples. Abbreviation: CAS, carotid atherosclerosis; PC, principal component.

leukocyte activation involved in immune response (Fig 5A). They were also enriched in cellular components such as the secretory granule membrane, external side of the plasma membrane, and tertiary granule (Fig 5A), as well as in molecular functions such as immune

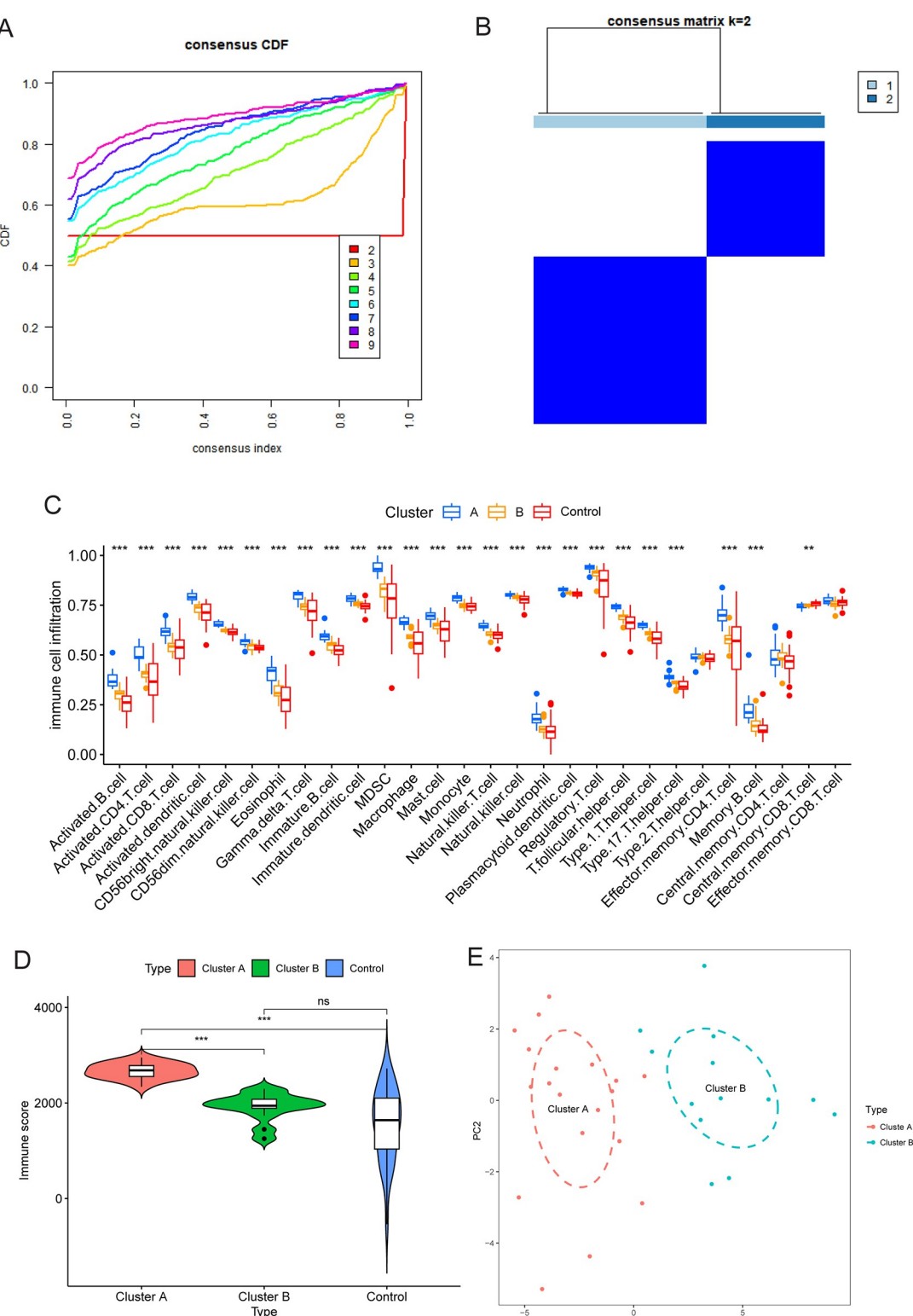

**Fig 3. High and low immune infiltration subtypes of CAS samples in the GSE43292 cohort distinguished via consensus clustering.** (A) Cumulative distribution function curve for different K values. (B) Consensus matrix when K = 2. CAS samples were divided into cluster A and cluster B. (C) Differences in the infiltration level of immune cell subtypes between cluster A, cluster B, and the control group. (D) Differences in the infiltration level of immune cell subtypes between cluster A, cluster B, and the control group. (E) Principal component analysis based on the infiltration level of immune cell subtypes in cluster A and

cluster B CAS samples. Abbreviation: CDF, cumulative distribution function; PC, principal component. Note: ***, P < 0.001; **, P < 0.01; ns, P > 0.05.

receptor activity (Fig 5A). Finally, they were enriched in KEGG pathways related to the immune response (Fig 5B). Conversely, up-regulated DEGs in the control group of the GSE43292 cohort were enriched in biological processes such as muscle system process, muscle contraction, and muscle tissue development (Fig 5C), in cellular components associated with contractile fiber, myofibril, and sarcomere (Fig 5C), and in molecular functions involving the structural constituent of muscle (Fig 5C). Notably, they were also enriched in KEGG pathways involving dilated cardiomyopathy, hypertrophic cardiomyopathy, and arrhythmogenic right ventricular cardiomyopathy (Fig 5D). Afterward, PPI networks were constructed based on the up-regulated and down-regulated genes of cluster A compared to the control samples. The PPI data of both up-regulated and down-regulated gene sets were imported into Cytoscape software, following which the maximum neighborhood algorithm was applied to identify the top 10 hub genes with the strongest interactions. The highest-ranking hub gene in the PPI network of the up-regulated gene set in cluster A was *PTPRC* (Fig 5E), while that of the down-regulated gene set was *ACTN2* (Fig 5F). Interestingly, the relationship between *ACTC1*, *LDB3*, *MYOZ2*, *TPM2*, *and CAS* has not been described in previous literature.

## Diagnostic efficacy of hub genes

The expression level of *PTPRC* was higher in the cluster A samples compared to the cluster B and control samples (Fig 6A), whereas that of *ACTN2* was lower (Fig 6B). The area under the ROC curve of *ACTN2* and *PTPRC* was 0.844 and 0.787 for distinguishing between CAS and control samples, respectively (Fig 6C and 6D). Moreover, the area under the ROC curve of *ACTN2* and *PTPRC* was 0.986 and 0.939 for distinguishing between cluster A and cluster B, respectively (Fig 6E and 6F). The expression level of *PTPRC* was higher in advanced CAS samples compared to early CAS samples (Fig 7A). In contrast, the expression level of *ACTN2* was lower in advanced CAS samples compared with early CAS samples (Fig 7B). The area under the ROC curve of *ACTN2* and *PTPRC* was 0.913 and 0.909 for differentiating between advanced CAS samples and early CAS samples, respectively (Fig 7C and 7D). In CAS samples of the GSE43292 and GSE28829 cohorts, *ACTN2* expression was negatively correlated with the infiltration level of most immune cell subtypes (Fig 8A and 8B), whereas *PTPRC* expression was positively correlated with the infiltration level of most immune cell subtypes (Fig 8A and 8B). Notably, the expression levels of *ACTC1*, *LDB3*, *MYOZ2*, and *TPM2* were lower in cluster A samples compared with cluster B and control samples (Fig 9A–9D). The area under the ROC curve of *ACTC1*, *LDB3*, *MYOZ2*, and *TPM2* for distinguishing CAS samples from control samples was 0.790, 0.807, 0.831, and 0.834, respectively (Fig 9E–9H). The area under the ROC curve of *ACTC1*, *LDB3*, *MYOZ2*, and *TPM2* for distinguishing between cluster A and cluster B was 0.927, 0.988, 0.960 and 0.992, respectively (Fig 9I–9L). Of note, the expression levels of *ACTC1* (Fig 10A), *MYOZ2* (Fig 10C), and *TPM2* (Fig 10D) were lower in advanced CAS samples compared with early CAS samples. However, LDB3 expression was comparable between advanced and early CAS samples (Fig 10B). The area under the ROC curve of *ACTC1*, *LDB3*, *MYOZ2*, and *TPM2* for differentiating advanced CAS samples from early CAS samples was 0.721, 0.683, 0.764, and 0.865, respectively (Fig 10E–10H). In the CAS samples of the GSE43292 (Fig 11A) and GSE28829 cohorts (Fig 11B), the expression of *ACTC1*, *LDB3*, *MYOZ2*, and *TPM2* was negatively correlated with infiltration levels of most immune cell subtypes.

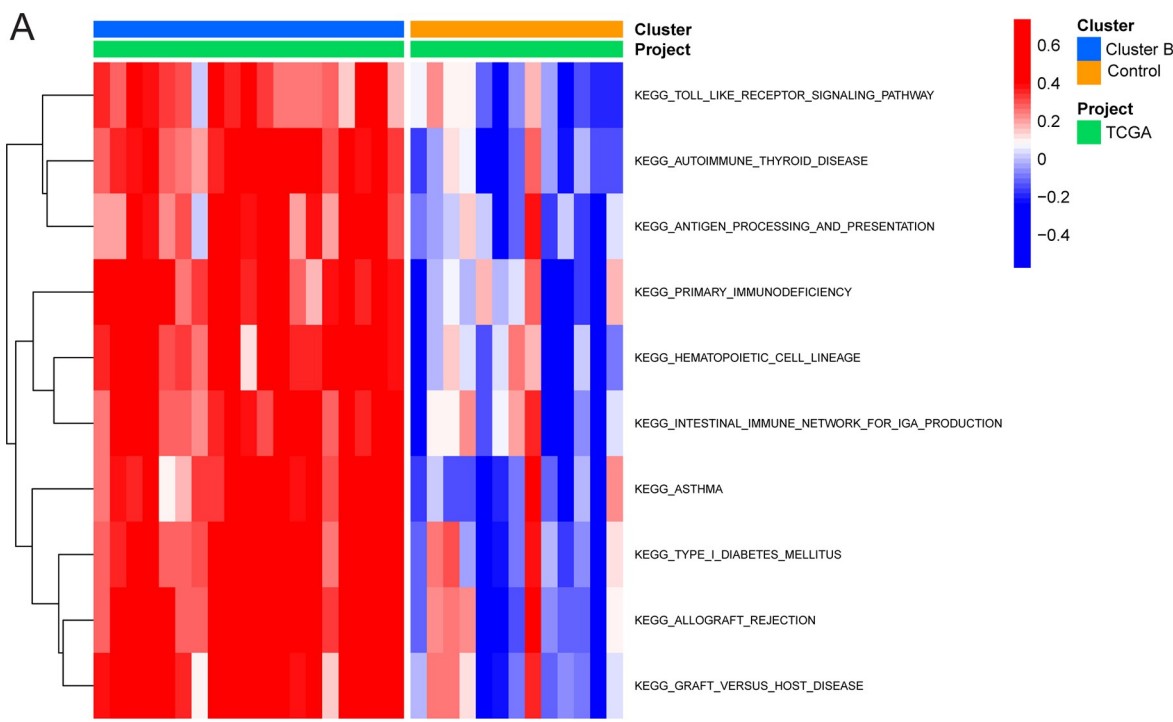

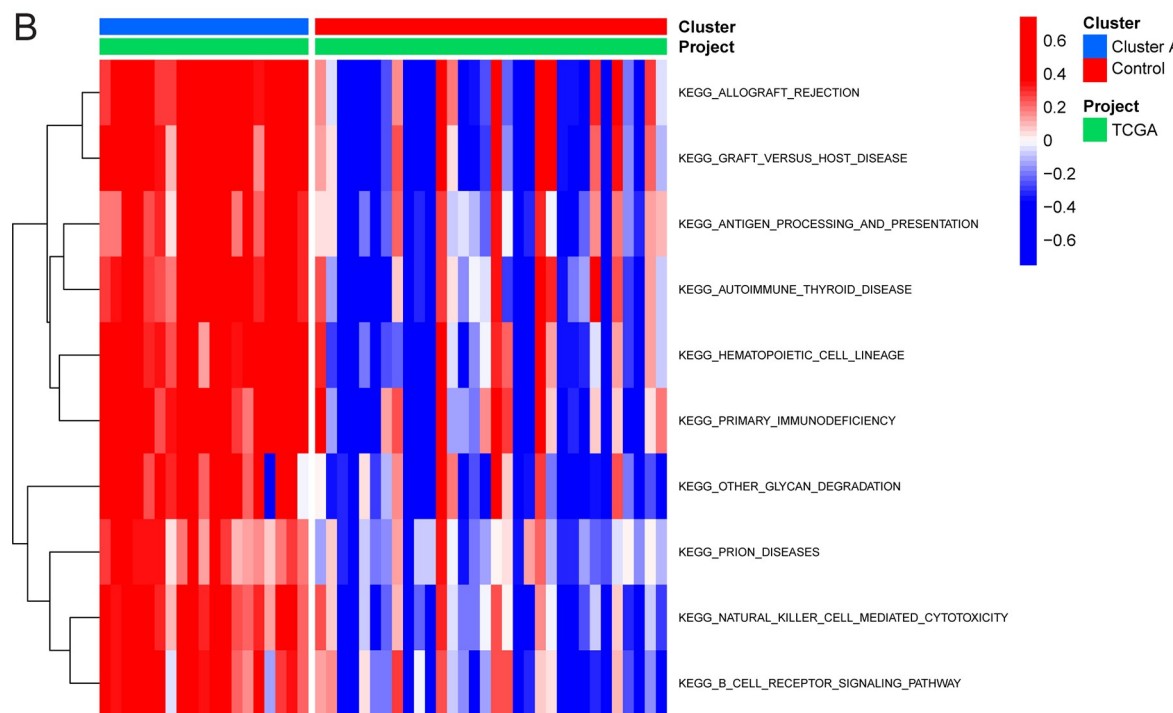

**Fig 4. Gene set variation analysis (GSVA) between cluster A, cluster B, and the control group.** (A) GSVA between cluster B and the control group. (B) GSVA between cluster A and the control group.

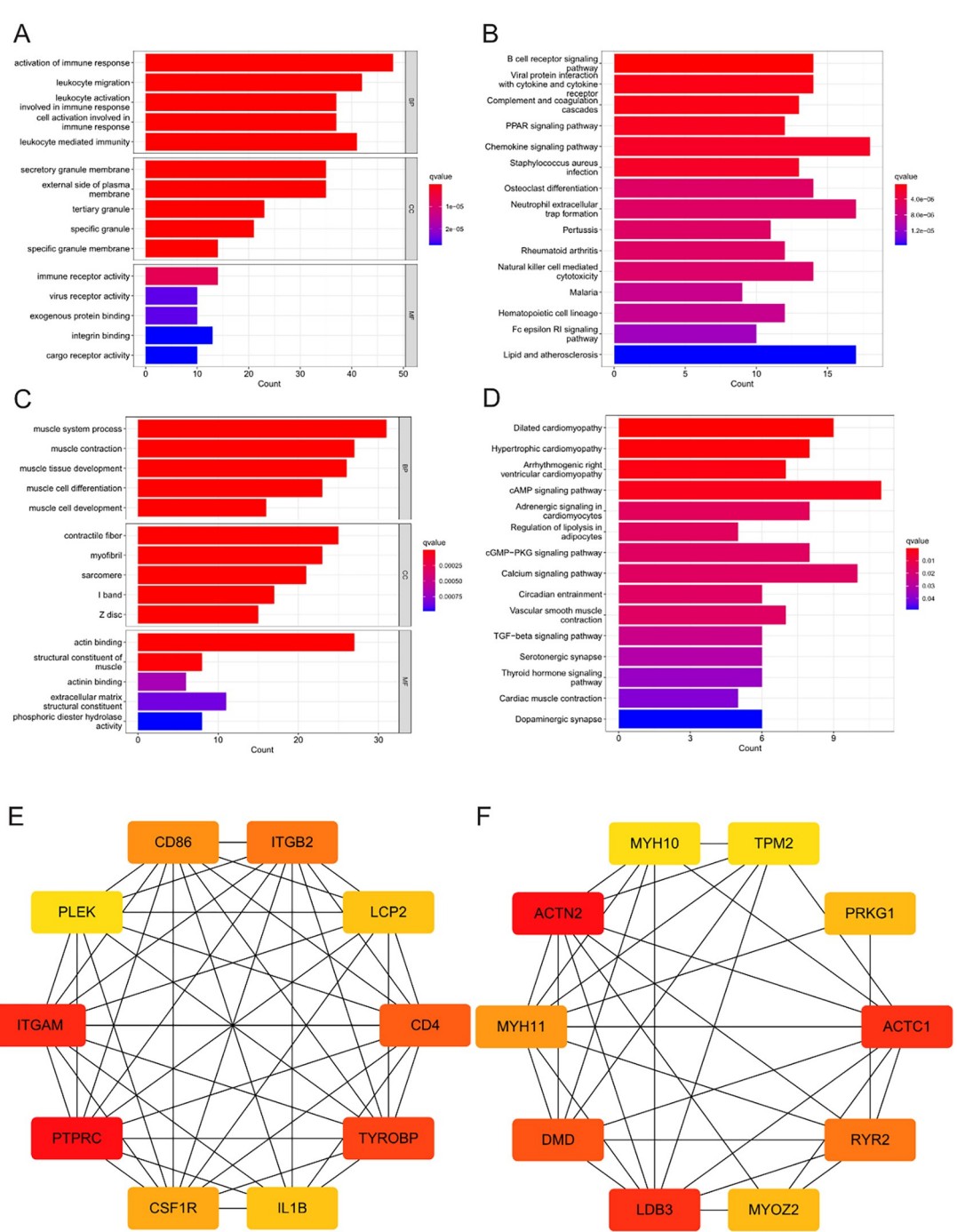

**Fig 5. Enrichment analysis and hub gene screening.** (A) GO enrichment analysis of up-regulated genes in cluster A compared to the control group. (B) KEGG enrichment analysis of up-regulated genes in cluster A compared to the control group. (C) GO enrichment analysis of down-regulated genes in cluster A compared to the control group. (D) KEGG enrichment analysis of down-regulated genes in cluster A compared to the control group. (E) Hub gene interaction network of up-regulated genes in cluster A compared to the control group. Red to yellow indicates that genes rank from high to low in the interaction network,

with *PTPRC* being the highest-ranking gene identified via the maximum neighborhood component algorithm. (F) Hub gene interaction network of down-regulated genes in cluster A compared to the control group. Red to yellow indicates that genes rank from high to low in the interaction network, with *ACTN2* being the highest-ranking gene identified via the maximum neighborhood component algorithm. Abbreviation: BP, Biological Process; CC, Cellular Component; MF, molecular function.

## Discussion

At present, advances in microarray have facilitated the comprehensive analysis of mRNA expression profiles across the global genome. The GSE43292 dataset contained the RNA sequencing data of CAS vascular tissue and adjacent healthy vascular tissue pairs, whilst the GSE28829 comprised early and advanced CAS vascular tissue. These two sets of samples, including control vascular tissue, early diseased vascular tissue, and advanced diseased vascular tissue, enabled the characterization of changes in the immune microenvironment throughout disease progression to determine trends and statuses of the immune microenvironment during pathological transformation associated with the disease.

The results of this study unveiled that compared with healthy vascular tissue, immune cells were enriched in CSA samples throughout disease progression. Specifically, the level of immune cell infiltration was higher in advanced CAS samples compared to early CAS samples. Previous studies reported that various immune cell subtypes play a pivotal role in the pathogenesis of CAS [24–26]. Macrophages drive atherosclerosis and associated complications, playing distinct roles across all stages of atherosclerosis, and are also the most abundant type of inflammatory cells in plaques [27]. In addition to the high level of infiltration of macrophages, their plasticity and heterogeneity play a vital role in the immune response to atherosclerosis [28]. Indeed, macrophages can shift their phenotype according to their microenvironment to allow them to fulfill specific roles [28, 29]. For instance, M1 macrophages promote and maintain inflammatory responses and thus exert atherogenic effects [30]. In contrast, type 2 T helper cells stimulate the differentiation of M2 macrophages [31, 32], which express high levels of anti-inflammatory factors, balance the activity of M1 macrophages, alleviate inflammatory responses, initiate tissue repair, promote tissue remodeling and angiogenesis, and exert phagocytic effects to mediate the inflammatory response, thereby exerting anti-atherosclerotic effects [33, 34]. Herein, the level of type 2 T helper cells remained unchanged in both early and advanced CAS samples. Dendritic cells play an essential role in CAS by stimulating chemokine and cytokine secretion and participating in antigen presentation and lipid uptake, which may induce inflammation or promote immune tolerance [35–37]. B cells may be implicated in systemic and local immune responses to atherosclerosis by modulating cellular immune responses through intercellular contact, antigen presentation, and cytokine production [38, 39]. T cells are the second largest group of immune cells in carotid atherosclerotic plaques after macrophages. Th1 cells were the dominant form in atherosclerotic plaques, forming a mutually stimulating loop with macrophages conducing to disease progression [40, 41].

Furthermore, PCA demonstrated that the level of immune cell infiltration could be used to distinguish healthy vascular tissue from CAS samples, both early and late stages. The immune microenvironment is one of the essential characteristics of disease and can be used to describe different disease phenotypes. The enrichment of immune cells in vascular tissues was identified as a principal driving force behind pathological changes in diseases. Characterizing the immune microenvironment characteristics of CAS samples and exploring the enrichment mechanism of immune cell subtypes in CAS samples may potentially assist in elucidating disease heterogeneity and the molecular mechanism underlying disease progression. Hub genes that regulate the immune microenvironment or are significantly associated with the infiltration of immune cell subtypes may be potential diagnostic markers and therapeutic targets.

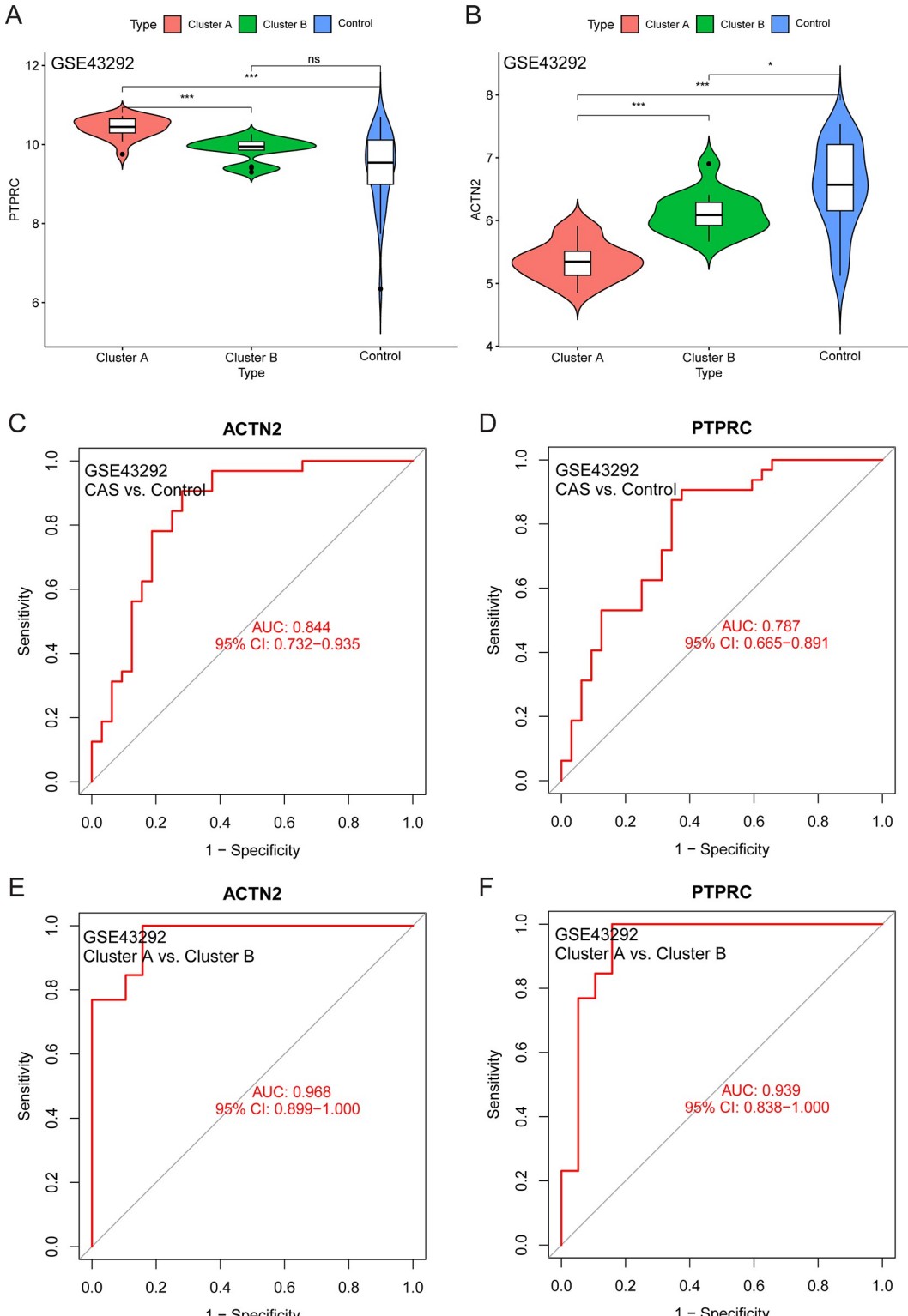

**Fig 6. Diagnostic efficiency evaluation of *PTPRC* and *ACTN2* in the GSE43292 cohort.** (A and B) Differences in *PTPRC* and *ACTN2* expression between cluster A, cluster B, and the control group. (C and D) *ACTN2* and *PTPRC* diagnostic efficiency on distinguishing the carotid atherosclerosis and control samples. (E and F) The accuracy of *ACTN2* and *PTPRC* in distinguishing cluster A and cluster B samples. Abbreviation: CAS, carotid atherosclerosis; PC, principal component; AUC, area under curve; CI, confidence interval. Note: ***, P < 0.001; *, P < 0.05; ns, P > 0.05.

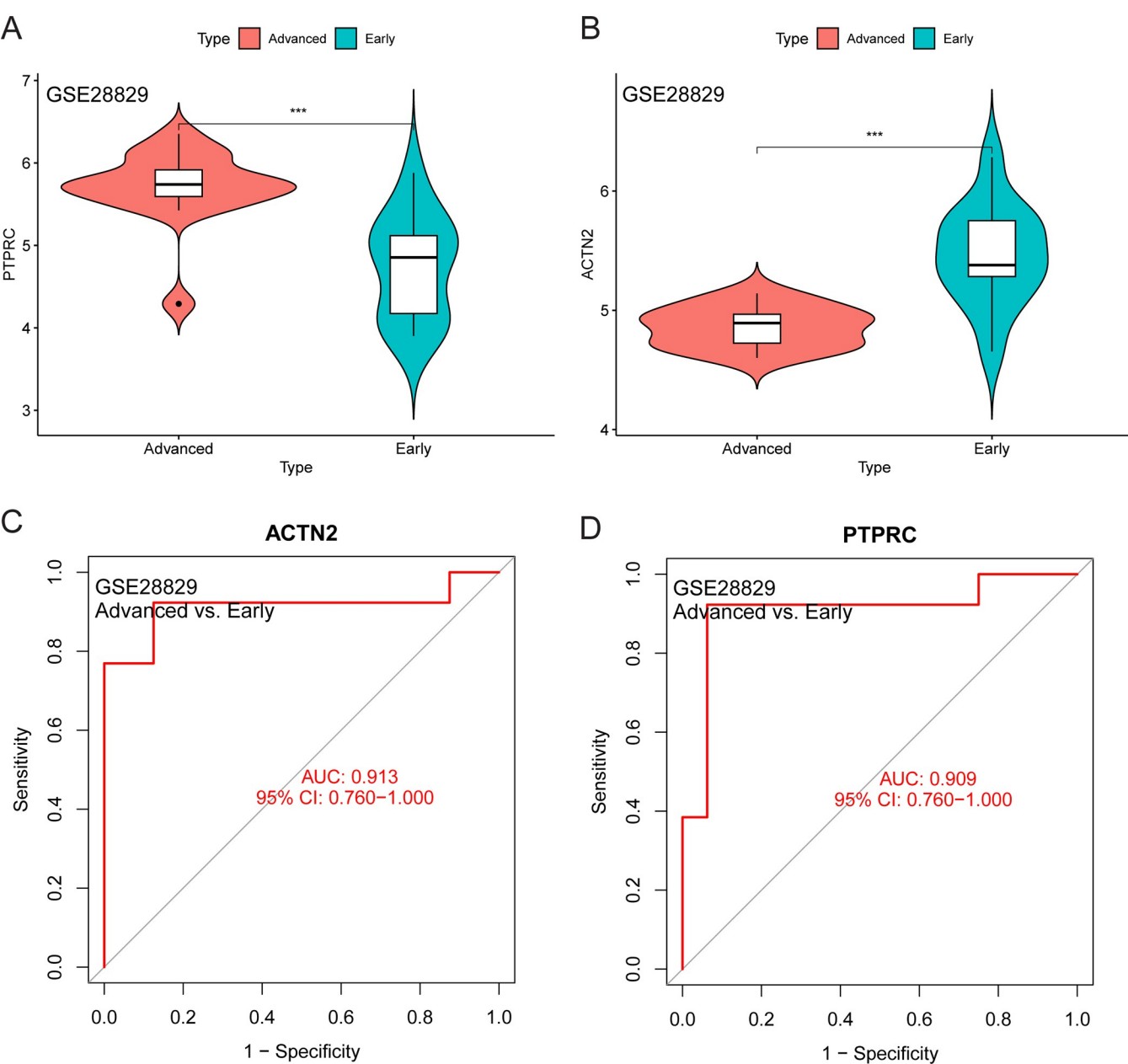

**Fig 7. Diagnostic efficiency validation of *PTPRC* and *ACTN2* in the GSE28829 cohort.** (A and B) Difference in *PTPRC* and *ACTN2* expression between advanced and early CAS samples. (C and D) The accuracy of *ACTN2* and *PTPRC* for distinguishing between advanced and early CAS samples. Abbreviation: CAS, carotid atherosclerosis; PC, principal component; AUC, area under curve; CI, confidence interval. Note: ***, P < 0.001; *, P < 0.05; ns, P > 0.05.

To delineate the heterogeneity of CAS samples and identify the enrichment mechanism of immune cells, CAS samples were divided into cluster A, hallmarked by a high infiltration level of immune cell subtypes, and cluster B, featuring a low infiltration level of immune cell sub-types via consensus clustering. In order to detect changes in the immune microenvironment, differentially expressed genes were identified between high immune infiltration clusters and healthy vascular tissues. Importantly, up-regulated genes in cluster A were significantly enriched in biological processes related to immune response, whereas down-regulated genes were significantly enriched in biological processes related to muscle development. The

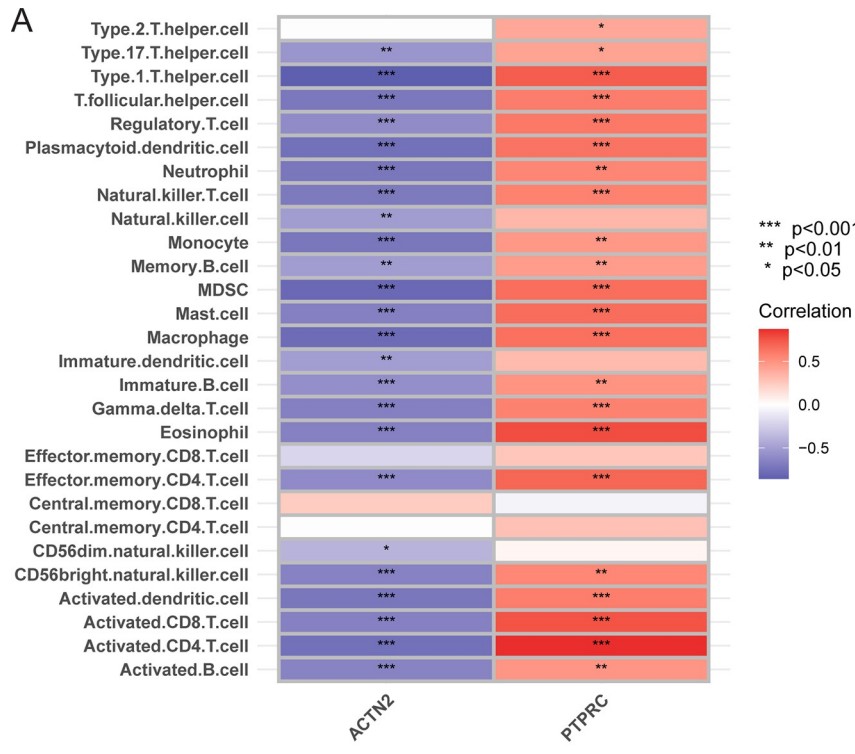

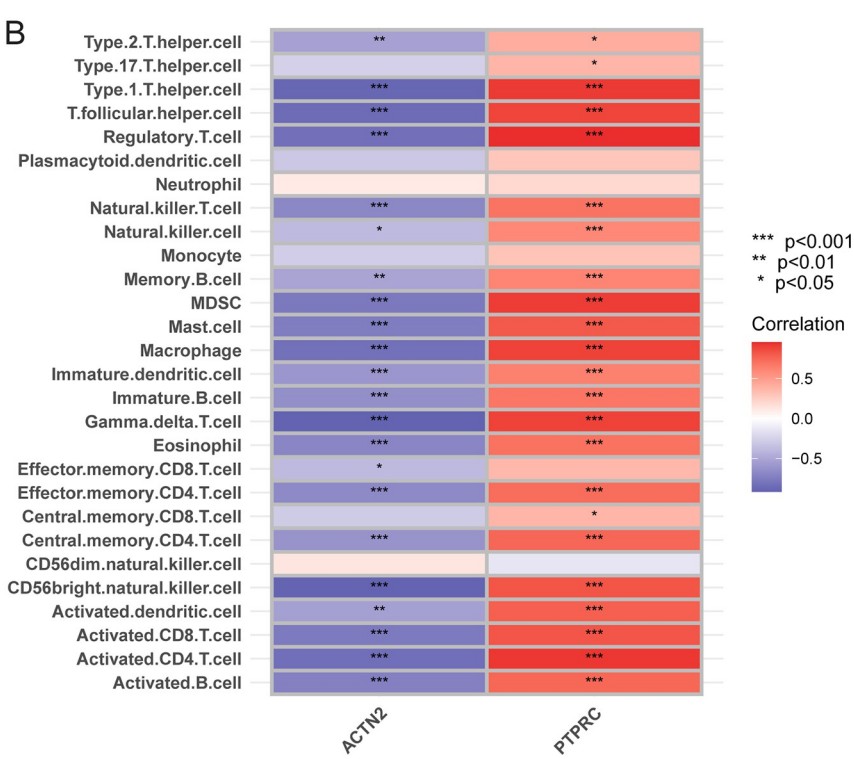

**Fig 8. Correlation between *PTPRC*, *ACTN2*, and immune cell subtypes.** (A) Correlation between *PTPRC*, *ACTN2*, and immune cell subtypes in CAS samples in the GSE43292 cohort. (B) Correlation between *PTPRC*, *ACTN2*, and immune cell subtypes in CAS samples in the GSE28829 cohort. Note: ***, P < 0.001; **, P < 0.01; *, P < 0.05.

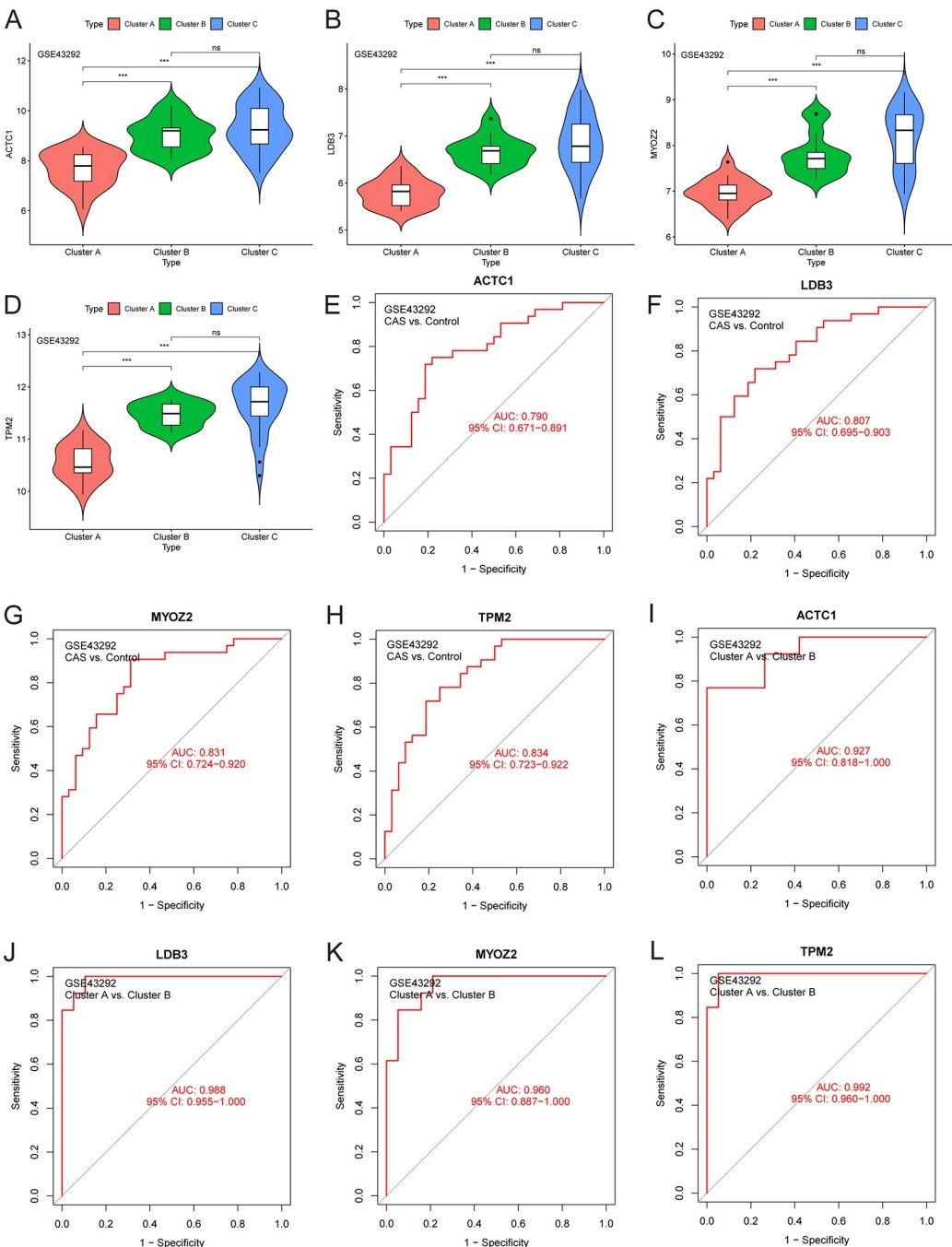

**Fig 9. Diagnostic efficiency evaluation of hub genes in the GSE43292 cohort.** (A-D) Differences in *ACTC1*, *LDB3*, *MYOZ2*, and *TPM2* expression between cluster A, cluster B, and the control group. (E-H) The accuracy of *ACTC1*, *LDB3*, *MYOZ2*, and *TPM2* for distinguishing between CAS and control samples. (I-L) The accuracy of *ACTC1*, *LDB3*, *MYOZ2*, and *TPM2* for distinguishing between cluster A and cluster B samples. Abbreviation: CAS, carotid atherosclerosis; AUC, the area under the curve; CI, confidence interval. Note: ***, P < 0.001; *, P < 0.05; ns, P > 0.05.

pathogenesis of atherosclerotic lesions has been hypothesized to be an exaggerated response to damage of endothelial cells and smooth muscle cells in the arterial wall [42, 43]. According to prior investigations, smooth muscle cell proliferation, smooth muscle cell synthesis, and

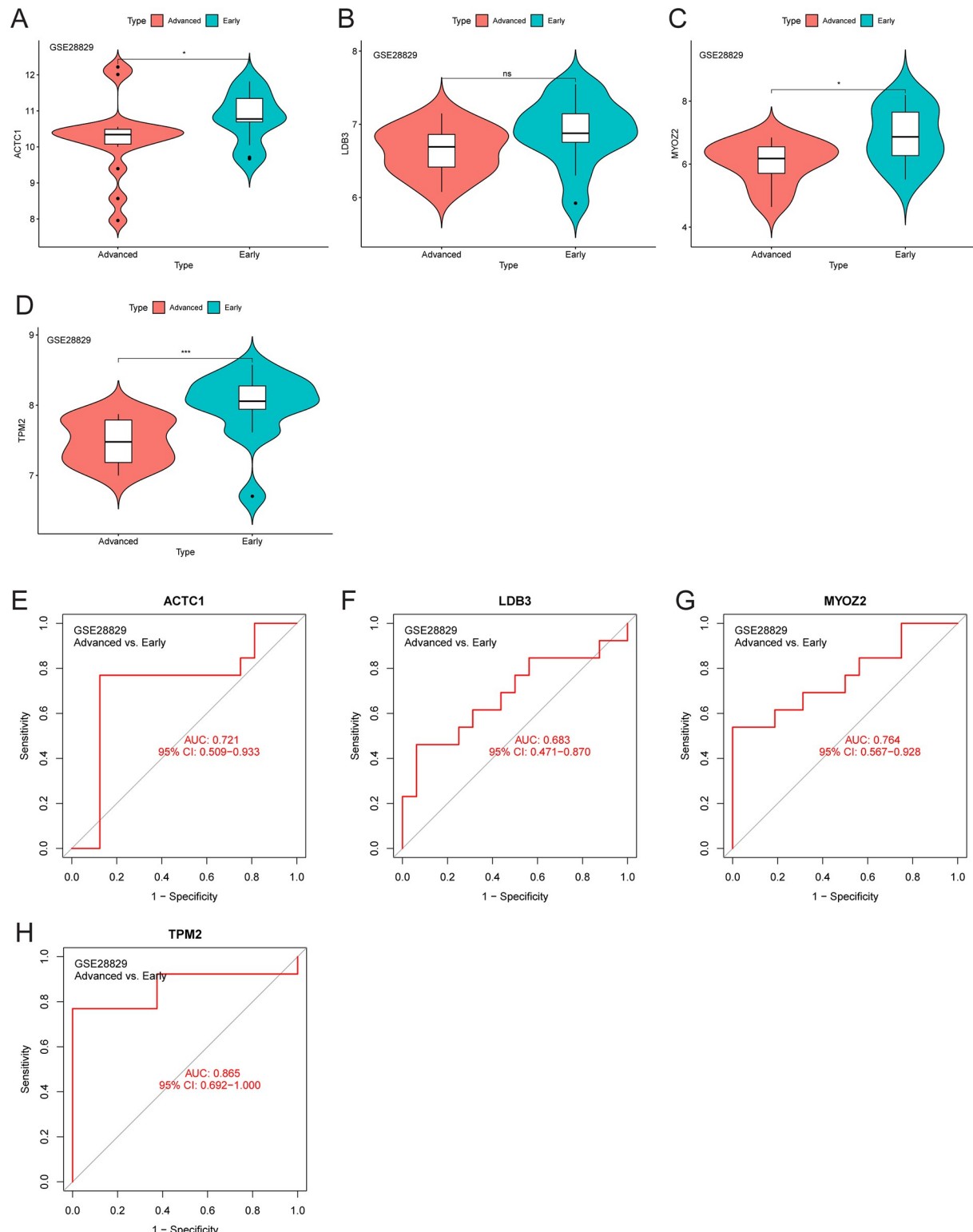

**Fig 10. Diagnostic efficiency validation of *ACTC1*, *LDB3*, *MYOZ2*, and *TPM2* in the GSE28829 cohort.** (A-D) Differences in *ACTC1*, *LDB3*, *MYOZ2*, and *TPM2* expression between advanced and early CAS samples. (E-H) The accuracy of *ACTC1*, *LDB3*, *MYOZ2*, and *TPM2* for distinguishing advanced from early CAS samples. Abbreviation: CAS, carotid atherosclerosis; AUC, the area under the curve; CI, confidence interval. Note: ***, P < 0.001; *, P < 0.05; ns, P > 0.05.

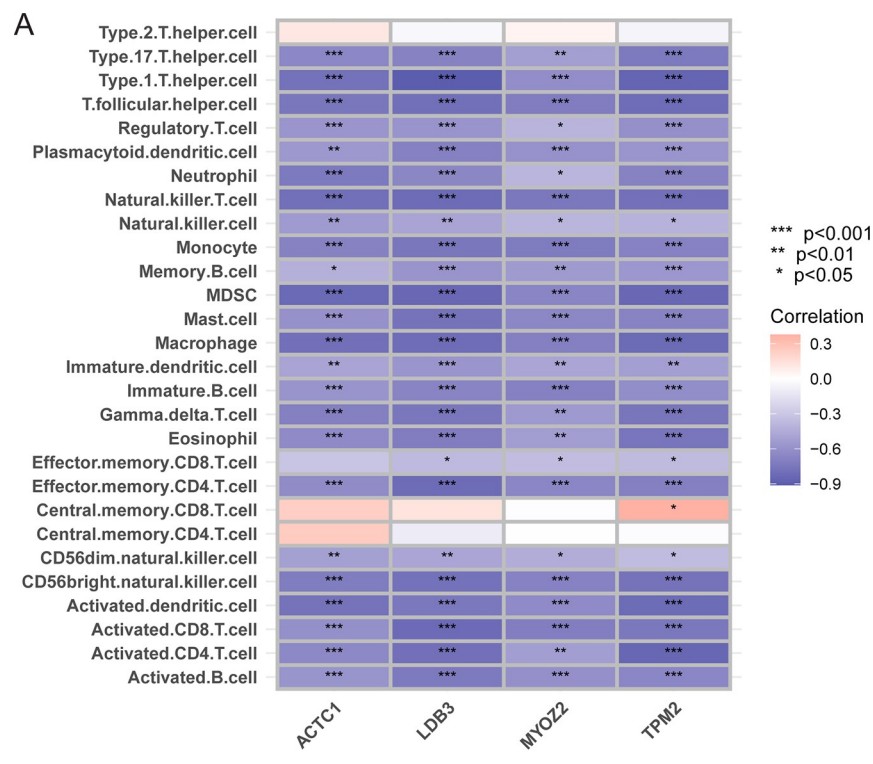

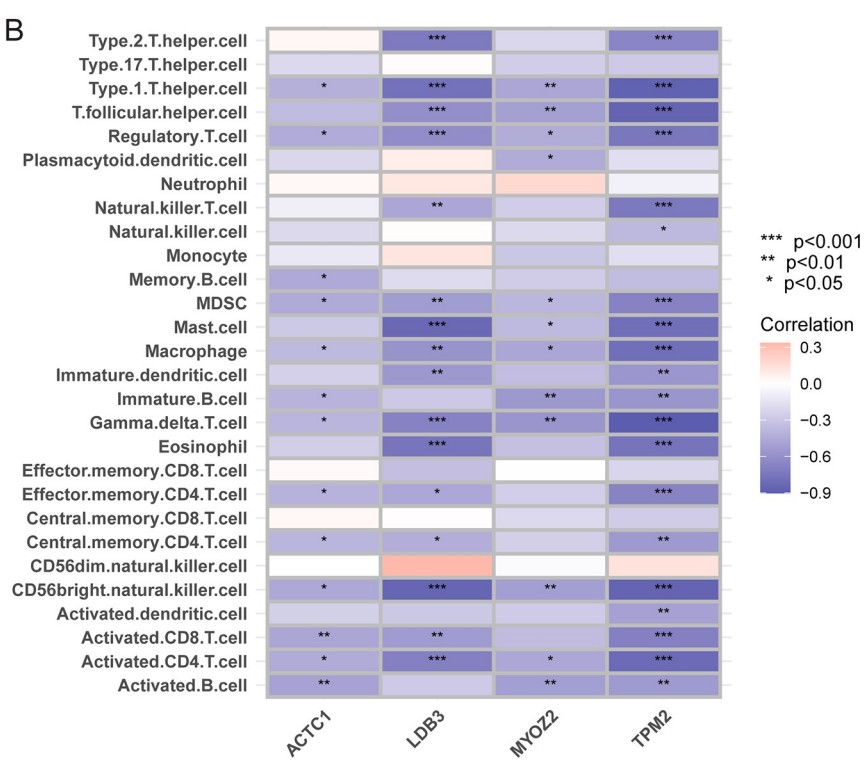

**Fig 11. Correlations between *ACTC1*, *LDB3*, *MYOZ2*, *TPM2*, and immune cell subtypes.** (A) Correlation between *ACTC1*, *LDB3*, *MYOZ2*, *TPM2*, and immune cell subtypes in CAS samples in the GSE43292 cohort. (B) Correlation between *ACTC1*, *LDB3*, *MYOZ2*, *TPM2*, and immune cell subtypes in CAS samples in the GSE28829 cohort. Note: ***, P < 0.001; **, P < 0.01; *, P < 0.05.

secretion of connective tissue components are the critical pathological changes observed in atherosclerosis [44, 45]. The results of the enrichment analysis validated the representativeness of the selected DEGs and the accuracy of sample classification.

To identify hub genes that regulate or are significantly associated with the immune micro-environment, PPI networks were generated, and the MNC algorithm was applied to identify the highest-ranking hub genes, namely *PTPRC* and *ACTN2*. These genes were effective in disease diagnosis and differentiation between early and advanced stages of the disease. Specifically, *PTPRC* expression was positively correlated with most immune cell subtypes, with its expression level increasing with disease progression, implying that it promoted the development of the disease.

In comparison, *ACTN2* expression was negatively correlated with most immune cell subtypes, and its expression level decreased with disease progression, suggesting that it impeded disease progression.

*PTPRC* is highly implicated in immune responses. It can promote T cell proliferation and differentiation, B cell proliferation, participate in T cell and B cell signal transduction pathways, positively regulate antigen receptor-mediated signal transduction pathways and protein kinases, release calcium ions into the cytoplasm, defend against responsive viruses, and regulate the cell cycle [46–48]. *ACTN2* is highly expressed in the cytoskeleton and actin filaments and can bind with actin and calcium ions to participate in the formation of muscle structure. Besides, it is a key molecule that maintains the physiological morphology and function of muscles. Down-regulating the expression of *ACTN2* may accelerate the loss of vascular smooth muscle function and elasticity. Nevertheless, studies examining the role of these two genes in carotid atherosclerosis are scarce, warranting further exploration [49, 50].

The relationship between *ACTC1*, *LDB3*, *MYOZ2*, *TPM2*, *and CAS* remains elusive. The expression levels of *ACTC1*, *LDB3*, *MYOZ2*, *and TPM2* were negatively correlated with the infiltration level of most immune cell subtypes, positioning them as potential protective factors against CAS. In addition, while these genes are associated with muscle development and physiological processes, their functions in CAS remain to be elucidated.

It is worthwhile recognizing that previous studies have used the same dataset as ours and identified hub genes that, while partially overlapping, are not completely consistent. This may be ascribed to inconsistencies in the algorithms used. For example, Liu et al. identified RBM47, HCK, CD53, TYROBP, and HAVCR2 as hub genes in advanced atherosclerotic plaques via network-based analysis [51]. They first used WGCNA to screen key modules, then constructed protein interaction networks to screen hub genes, all of which were pathogenic genes. Our study initially distinguished the immune phenotypes of the samples using consensus clustering and subsequently screened differentially expressed genes using the maximum neighborhood component algorithm. These hub genes may play an instrumental role in the differentiation of immune phenotypes among samples. Herein, hub genes were screened from two perspectives of high and low immune cell infiltration, including potential pathogenic and protective genes.

However, some limitations of our study should not be overlooked. To begin, the screening models for diagnostic genes are diverse and complex, and the diagnostic efficacy of the screened genes remains unknown in other datasets. Thus, high-quality cohorts with sufficient sample sizes are necessary to validate our results. Secondly, regularization techniques and cross-validation were not utilized in the present study. Applying these methods through various machine learning algorithms can enhance the accuracy and reliability of results. Considering the limited number of genes screened using the maximum neighborhood component algorithm, some genes might have been validated by previous studies. Therefore, instead of further narrowing our scope with regularization techniques and cross-validation, the diagnostic efficacy of all unpublished genes in the downloaded datasets was validated.

## Conclusion

The enrichment of immune cells in vascular tissues promoted pathological changes in CAS. Advanced CAS was characterized by high immune cell infiltration levels, whereas early CAS was hallmarked by low immune cell infiltration levels. Moreover, CAS progression may be related to the immune response pathway. Biological processes related to muscle cell development may delay CAS progression. Meanwhile, the expression of the hub genes CAS. *PTPRC* and *ACTN2*, *ACTC1*, *LDB3*, *MYOZ2, and TPM2*, identified in the interaction network of differentially expressed genes between cluster A and healthy vascular tissues, play an essential role in regulating the immune microenvironment of CAS and participate in its occurrence and progression. Finally, *PTPRC* and *ACTN2*, *ACTC1*, *LDB3*, *MYOZ2, and TPM2* demonstrated favorable efficacy for distinguishing between high and low immune cell infiltration CAS samples, as well as for differentiating early from advanced CAS stages.

## Author Contributions

**Conceptualization:** Xianming Hou.

**Data curation:** Yi Zhang, Lingmin Zhang, Jing Fang.

**Investigation:** Xianming Hou.

**Methodology:** Xianming Hou.

**Software:** Yi Zhang, Lingmin Zhang.

**Writing – original draft:** Yi Zhang, Lingmin Zhang, Yunfang Jia, Xianming Hou.

**Writing – review & editing:** Yi Zhang, Yunfang Jia, Jing Fang, Shuancheng Zhang.

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
