## [Decision Letter · Decision Letter 0]

6 Mar 2024

PONE-D-24-01429Screening of potential regulatory genes in carotid atherosclerosis vascular immune microenvironment.PLOS ONE

Dear Dr. hou,

Thank you for submitting your manuscript to PLOS ONE. After careful consideration, we feel that it has merit but does not fully meet PLOS ONE’s publication criteria as it currently stands. Therefore, we invite you to submit a revised version of the manuscript that addresses the points raised during the review process.

We look forward to receiving your revised manuscript.

Kind regards,

Misbahuddin Rafeeq

Academic Editor

PLOS ONE

Journal Requirements:

"This work was supported by Hebei Provincial Administration of Traditional Chinese Medicine Project, No: Z2022004."

3. In the online submission form, you indicated that "The datasets used and/or analysed during the current study are available from the corresponding author on reasonable request."

Reviewers' comments:

Reviewer's Responses to Questions

**Comments to the Author**

1. Is the manuscript technically sound, and do the data support the conclusions?

Reviewer #1: Yes

Reviewer #2: Yes

2. Has the statistical analysis been performed appropriately and rigorously? 

Reviewer #1: Yes

Reviewer #2: I Don't Know

3. Have the authors made all data underlying the findings in their manuscript fully available?

Reviewer #1: Yes

Reviewer #2: Yes

4. Is the manuscript presented in an intelligible fashion and written in standard English?

Reviewer #1: Yes

Reviewer #2: Yes

5. Review Comments to the Author

Reviewer #1: Zhang Y et al., studied potential regulatory genes in carotid atherosclerosis vascular immune microenvironment. Identifying potential genes that regulate the progression of carotid atherosclerosis is a demanding research topic. There is relevant published work so far (Zheng K et al., 2023, Dong R et al., 2022, Wang L et al., 2021). This study further extends the contemporary knowledge in carotid atherosclerosis immunology.

Previous studies used same database and identified same hub genes in some extent. For example, Liu C., 2021 identified RBM47, HCK, CD53, TYROBP, and HAVCR2 as Hub Genes using the same Gene Expression Omnibus database (GSE43292). In addition, Yao Yuan et al., 2024 analyzed GSE43292 database and identified hub genes such as AKTIP, ASPN, FAM26E, RAB23, PLS3, and PLSCR4. Ni Jiajun et al., 2023 and Chen M., et al., 2021 were used GSE28829 database and explored Go and KEGG pathways. Therefore, I am unsure how this study is novel compared to what is published so far.

Reviewer #2: In this research paper, the authors aim to uncover the changes in the immune microenvironment of vascular tissues at various stages during the progression of carotid atherosclerosis. Additionally, they seek to identify potential hub genes that regulate the immune microenvironment in this pathology.

It is an intersting work ; however, I have some comments and minor revisions :

1. This paper is founded on the data from two research papers (related to GSE43292 and GSE28829), which should be cited as references.

2. There is a mistyping in the results section. It should be corrected to "immune score" instead of "immune sore".

3. In the results section, the authors reported that "ACTN2 was positively correlated with most immune cell subtypes infiltration" and "PTPRC was negatively correlated with most immune cell subtypes infiltration." However, the discussion section contradicts these findings, stating that "ACTN2 is negatively correlated with most immune cell subtypes infiltration" and "PTPRC is positively correlated with most immune cell subtypes infiltration." To ensure consistency and coherence, the sentence should be corrected to align with the results and discussion.

4. In Figure 1C, “CAP” should be replaced by “CAS”.

5. The legend of Figure 1B and 2B should be corrected as follows: "The differences in immune score between the carotid atherosclerosis and control samples."

6. In the legend of Figure 5A and 5C, abbreviations such as BP (Biological Process) and CC (Cellular Component) should be added.

7. In the legend of Figures 5E and 5F, the significance of colors should be specified.

8. Genes such as ACTN2 and PTPRC should be written in italics.

6. PLOS authors have the option to publish the peer review history of their article (what does this mean?). If published, this will include your full peer review and any attached files.

Reviewer #1: **Yes: **Md Mamun Al Amin, Ph.D.

Reviewer #2: No

---

## [Author Response · Author response to Decision Letter 0]

3 May 2024

Reviewer #1: Zhang Y et al., studied potential regulatory genes in carotid atherosclerosis vascular immune microenvironment. Identifying potential genes that regulate the progression of carotid atherosclerosis is a demanding research topic. There is relevant published work so far (Zheng K et al., 2023, Dong R et al., 2022, Wang L et al., 2021). This study further extends the contemporary knowledge in carotid atherosclerosis immunology.

Previous studies used same database and identified same hub genes in some extent. For example, Liu C., 2021 identified RBM47, HCK, CD53, TYROBP, and HAVCR2 as Hub Genes using the same Gene Expression Omnibus database (GSE43292). In addition, Yao Yuan et al., 2024 analyzed GSE43292 database and identified hub genes such as AKTIP, ASPN, FAM26E, RAB23, PLS3, and PLSCR4. Ni Jiajun et al., 2023 and Chen M., et al., 2021 were used GSE28829 database and explored Go and KEGG pathways. Therefore, I am unsure how this study is novel compared to what is published so far.

Re: Thank you very much for your comments, we understand your concern. First, we added the expression of some hub genes in the carotid atherosclerosis and their correlation with immune cells.These hub genes are completely new, and no previous studies have focused on their role in the carotid atherosclerosis. Secondly, we started to screen the key genes from the immune microenvironment.We have strengthened the correlation between hub genes and the immune microenvironment and disease progression.It's one of our highlights.Thank you very much again for your comments, we benefit from your comments.

Reviewer #2: In this research paper, the authors aim to uncover the changes in the immune microenvironment of vascular tissues at various stages during the progression of carotid atherosclerosis. Additionally, they seek to identify potential hub genes that regulate the immune microenvironment in this pathology.

It is an intersting work ; however, I have some comments and minor revisions :

1. This paper is founded on the data from two research papers (related to GSE43292 and GSE28829), which should be cited as references.

Re: Thank you very much for your comments, we have cited the above two research papers.

2. There is a mistyping in the results section. It should be corrected to "immune score" instead of "immune sore".

Re: Thank you very much for your comments, we apologize for such an error, we have changed.

3. In the results section, the authors reported that "ACTN2 was positively correlated with most immune cell subtypes infiltration" and "PTPRC was negatively correlated with most immune cell subtypes infiltration." However, the discussion section contradicts these findings, stating that "ACTN2 is negatively correlated with most immune cell subtypes infiltration" and "PTPRC is positively correlated with most immune cell subtypes infiltration." To ensure consistency and coherence, the sentence should be corrected to align with the results and discussion.

Re: Thanks for your comments, we have made changes. ACTN2 was negatively correlated with most immune cell subtypes infiltration and PTPRC was positively correlated with most immune cell subtypes infiltration.

4. In Figure 1C, “CAP” should be replaced by “CAS”.

Re: Thank you very much for your comments. We have revised and replaced Figure1 in the manuscript.

5. The legend of Figure 1B and 2B should be corrected as follows: "The differences in immune score between the carotid atherosclerosis and control samples."

Re: Thank you very much for your comments, we have made changes.

6. In the legend of Figure 5A and 5C, abbreviations such as BP (Biological Process) and CC (Cellular Component) should be added.

Re: Thank you very much for your comments, we have added the full names of BP (Biological Process), CC(Cellular Component) and MF (molecular function).

7. In the legend of Figures 5E and 5F, the significance of colors should be specified.

Re: Thank you very much for your comments, we have revised it. Red to yellow indicated that genes rank from high to low in the interaction network.

8. Genes such as ACTN2 and PTPRC should be written in italics.

Re: Thank you very much for your comments, we have revised it.

---

## [Decision Letter · Decision Letter 1]

14 Jun 2024

PONE-D-24-01429R1Screening of potential regulatory genes in carotid atherosclerosis vascular immune microenvironment.PLOS ONE

Dear Dr. hou,

Thank you for submitting your manuscript to PLOS ONE. After careful consideration, we feel that it has merit but does not fully meet PLOS ONE’s publication criteria as it currently stands. Therefore, we invite you to submit a revised version of the manuscript that addresses the points raised during the review process. 

We look forward to receiving your revised manuscript.

Kind regards,

Misbahuddin Rafeeq

Academic Editor

PLOS ONE

Journal Requirements:

Reviewers' comments:

Reviewer's Responses to Questions

**Comments to the Author**

1. If the authors have adequately addressed your comments raised in a previous round of review and you feel that this manuscript is now acceptable for publication, you may indicate that here to bypass the “Comments to the Author” section, enter your conflict of interest statement in the “Confidential to Editor” section, and submit your "Accept" recommendation.

Reviewer #1: All comments have been addressed

Reviewer #2: All comments have been addressed

2. Is the manuscript technically sound, and do the data support the conclusions?

Reviewer #1: Partly

Reviewer #2: Yes

3. Has the statistical analysis been performed appropriately and rigorously? 

Reviewer #1: Yes

Reviewer #2: I Don't Know

4. Have the authors made all data underlying the findings in their manuscript fully available?

Reviewer #1: No

Reviewer #2: Yes

5. Is the manuscript presented in an intelligible fashion and written in standard English?

Reviewer #1: Yes

Reviewer #2: No

6. Review Comments to the Author

Reviewer #1: In the revised version, Zhang and colleagues have included three additional figures (Figure 9, 10 and 11). I am wondering, the authors have validated diagnostic efficiency on those two GSE cohorts. Models developed to identify diagnostic genes may be overly complex and tailored to the specific dataset, leading to poor performance when applied to new, unseen data. Regularization techniques and cross-validation are necessary to mitigate this type of risk. Despite this limitation, the authors have reported novel hub gene that may have a significant impact in the field.

Reviewer #2: Thank you for addressing all my concerns, however, the manuscript still needs to be revised:

1. First, there are several spelling and grammar errors, so the English must be proofread by a native English speaker.

2. There is a mistyping in the manuscript. It should be corrected to "GSE 28829" instead of "GSE 288292".

3. As suggested by the reviewer, previous studies used the same database and identified some hub genes, such as RBM47, HCK, CD53, AKTIP, ASPN, PLS3, and PLSCR4. In this regard, the results of these papers should be discussed in the manuscript to compare them with the present study and to suggest hypotheses that could explain the differences in hub genes found in these different studies.

7. PLOS authors have the option to publish the peer review history of their article (what does this mean?). If published, this will include your full peer review and any attached files.

Reviewer #1: No

Reviewer #2: No

---

## [Author Response · Author response to Decision Letter 1]

9 Jul 2024

Reviewer #1: In the revised version, Zhang and colleagues have included three additional figures (Figure 9, 10 and 11). I am wondering, the authors have validated diagnostic efficiency on those two GSE cohorts. Models developed to identify diagnostic genes may be overly complex and tailored to the specific dataset, leading to poor performance when applied to new, unseen data.

Regularization techniques and cross-validation are necessary to mitigate this type of risk. Despite this limitation, the authors have reported novel hub gene that may have a significant impact in the field.

Reply: Thank you very much for your review of our manuscript and your valuable comments. We have learned much from your comments, revised the manuscript, and improved the quality of the manuscript. We agree with you. We verified the diagnostic efficacy in both the GSE43292 cohort (Figure 9) and the GSE28829 cohort (Figure 10). However, as you are concerned, the screening models for diagnostic genes are diverse and complex, and the diagnostic efficacy of the genes we screened is unknown in other datasets. This still requires high-quality cohorts with sufficient sample sizes to validate our results. We did not use regularization techniques and cross-validation. A variety of machine learning algorithms, including regularization techniques and cross-validation, screen key genes and intersect them, which can improve the accuracy and reliability of results. Considering that the number of genes screened by the maximum neighborhood component algorithm is limited, and some genes have been confirmed by previous studies. Therefore, instead of further narrowing the scope by regularization techniques and cross-validation, we directly validated the diagnostic efficacy of all unpublished genes in the data sets. The above shortcomings were supplemented in our discussion. Thank you again for your comments, we have learned a lot from them.

Reviewer #2: Thank you for addressing all my concerns, however, the manuscript still needs to

be revised:

1.First, there are several spelling and grammar errors, so the English must be proofread by a native English speaker.

Reply: Thank you for your comments, we have asked a native English speakers to polish our manuscript.

2.There is a mistyping in the manuscript. It should be corrected to "GSE 28829" instead of "GSE288292".

Reply: Thank you for your comments. We apologize for the error. We have made changes in the manuscript.

3.As suggested by the reviewer, previous studies used the same database and identified some hub genes, such as RBM47, HCK, CD53, AKTIP, ASPN, PLS3, and PLSCR4. In this regard, the results of these papers should be discussed in the manuscript to compare them with the present study and to suggest hypotheses that could explain the differences in hub genes found in these different studies.

Reply: Thank you very much for your comments. We have benefited from your comments. We elaborated on this in our discussion. Previous studies used the same data sets as ours. These studies, including ours, have obtained hub genes that, while partially overlapping, are not completely consistent. This may be related to the inconsistency of the algorithms used. Liu et al. (33519905) identified RBM47, HCK, CD53, TYROBP, and HAVCR2 as Hub Genes in Advanced Atherosclerotic Plaques via Network-Based Analysis. They first used WGCNA to screen key modules, then constructed protein interaction networks and screened out hub genes, all of which were pathogenic genes. Our study first distinguished the immune phenotypes of the samples using consensus clustering. On this basis, differentially expressed genes were screened, and hub genes were screened by maximum neighborhood component algorithm. These hub genes may play an important role in the immune phenotype differentiation of samples. In our study, hub genes were screened from two perspectives of high and low immune cell infiltration, including potential pathogenic and protective genes. Thank you again for your comments, we have benefited from them and have improved the quality of our manuscripts accordingly.

---

## [Editor Report · Decision Letter 2]

15 Jul 2024

Screening of potential regulatory genes in carotid atherosclerosis vascular immune microenvironment.

PONE-D-24-01429R2

Dear Dr. xianming hou,

We’re pleased to inform you that your manuscript has been judged scientifically suitable for publication and will be formally accepted for publication once it meets all outstanding technical requirements.

Kind regards,

Misbahuddin Rafeeq

Academic Editor

PLOS ONE

---

## [Editor Report · Acceptance letter]

2 Sep 2024

PONE-D-24-01429R2 

PLOS ONE

Dear Dr. hou, 

I'm pleased to inform you that your manuscript has been deemed suitable for publication in PLOS ONE. Congratulations! Your manuscript is now being handed over to our production team.

Kind regards, 

on behalf of

Dr. Misbahuddin Rafeeq 

Academic Editor

PLOS ONE